# Blood–Brain Barrier Solute Carrier Transporters and Motor Neuron Disease

**DOI:** 10.3390/pharmaceutics14102167

**Published:** 2022-10-11

**Authors:** Sana Latif, Young-Sook Kang

**Affiliations:** Research Institute of Pharmaceutical Sciences, College of Pharmacy, Sookmyung Women’s University, 100 Cheongpa-ro 47-gil, Yongsan-gu, Seoul 04310, Korea

**Keywords:** solute carrier (SLC) transporters, blood–brain barrier (BBB), amyotrophic lateral sclerosis (ALS), NSC-34 cell lines, taurine transporter (Taut), large amino acid transporter 1 (LAT1), monocarboxylate transporters (MCTs), organic cation transporters (OCTNs), choline transporter-like protein-1 (CTL1)

## Abstract

Defective solute carrier (SLC) transporters are responsible for neurotransmitter dysregulation, resulting in neurodegenerative diseases such as amyotrophic lateral sclerosis (ALS). We provided the role and kinetic parameters of transporters such as ASCTs, Taut, LAT1, CAT1, MCTs, OCTNs, CHT, and CTL1, which are mainly responsible for the transport of essential nutrients, acidic, and basic drugs in blood–brain barrier (BBB) and motor neuron disease. The affinity for LAT1 was higher in the BBB than in the ALS model cell line, whereas the capacity was higher in the NSC-34 cell lines than in the BBB. Affinity for MCTs was lower in the BBB than in the NSC-34 cell lines. CHT in BBB showed two affinity sites, whereas no expression was observed in ALS cell lines. CTL1 was the main transporter for choline in ALS cell lines. The half maximal inhibitory concentration (IC_50_) analysis of [^3^H]choline uptake indicated that choline is sensitive in TR-BBB cells, whereas amiloride is most sensitive in ALS cell lines. Knowledge of the transport systems in the BBB and motor neurons will help to deliver drugs to the brain and develop the therapeutic strategy for treating CNS and neurological diseases.

## 1. Introduction

The solute carrier (SLC) superfamily constitutes more than 65 families and over 400 genes responsible for the influx and efflux of a wide range of molecules such as organic and inorganic ions, sugars, and amino acids across membranes [1]. These transporters are mainly facilitative or depend on ion gradient for the transport of substrates [2]. Specifically for substrates like amino acids, which are building blocks for proteins, the main transporters involved belong to SLC1, 3 SLC6, 7, and SLC25, 36 subfamilies [3]. The range of specificity differs even within the family [4], and mutations in about 71 SLC genes are related to brain diseases. Various SLC transporters have contributed to the identification of diseases and participate in the specific delivery of drugs and are therefore focused as the major targets for drug delivery in the treatment of diseases [5]. Brain homeostasis is maintained with the aid of the blood–brain barrier (BBB) and cerebrospinal fluid (CSF). The BBB has a complex structure that is made of endothelial cells with tight junctions. Brain capillaries are responsible for regulating the transport of metabolites and nutrients across the BBB [6]. About 287 SLC genes have been identified in the brain [7]. SLCs expressed in the BBB [8] protect the brain from toxins and aid the absorption of essential nutrients from the blood [9]. In addition, SLCs present in the glia and neurons play important roles in regulating drug response and brain homeostasis [8]. Hence, attention should be focused on targeting SLCs for treating brain diseases by targeting the modulation of SLCs for drug transport, specifically the movement of prodrugs and drugs from the blood to the brain [10]. Neurodegeneration is a major disorder caused by various factors, including genetic and environmental aspects such as nutrients [11]. Alteration in SLC polymorphism results in neurodegeneration by irregularities in the expression of transporters and abnormal neurotransmission. The SLC families are important targets for the therapeutic drugs used to treat CNS diseases [12]. Dysfunctions of neurotransmitters are mainly involved in neurodegenerative diseases. For instance, glutamate in amyotrophic lateral sclerosis (ALS), gamma-aminobutyric acid (GABA) in schizophrenia and epilepsy, and serotonin in Parkinson’s disease (PD).

Among motor neuron diseases, ALS is considered to be the most prevalent disorder resulting in muscle paralysis [13]. ALS cases are mostly sporadic, accounting for 90%, whereas familial cases are about 5–10% [14]. Mutation in the superoxide dismutase -1 gene (SOD1), TAR-DNA binding, chromosome 9 open reading frame 72, and fused sarcoma are the main causes of familial ALS [15]. Pathophysiological processes in ALS involve glutamate excitotoxicity, mitochondrial and axonal transport dysfunction, and increased oxidative stress. For the treatment of ALS, riluzole, an FDA-approved drug, works by reducing glutamate release, which is proposed to be mediated by an increase in the transporter SLC1A3, which removes glutamate from synapses, resulting in decreased glutamate levels [16,17]. In addition to this, in astroglial cells, riluzole has been shown to enhance the uptake of glutamate through elevating the SLC1A1 levels, which is an excitatory amino acid transporter [12]. Further earlier reports have shown that glutamate levels were decreased via decrease in the glutamate receptors GLT1 isoform in sensory and motor cortex in sporadic ALS patients [18]. A member of mitochondrial carrier family SLC25A20 has a role in transportation across the mitochondrial inner membrane, and the possible mechanisms in ALS are reported as maintaining calcium homeostasis, ATP production and mitochondrial apoptosis regulation [19].

The exact mechanism of the ALS is unknown, therefore a set of in vitro and in vivo experimental models are being used to validate how the mutation in the SOD1 gene leads to injury of the motor neuron [20]. In ALS mice, altered levels of amino acids have been shown in the spinal cord (lumber) in comparison with the control type (WT) mice [21]. Therefore, we conducted a series of research works to find the alteration in the transport of amino acids and acidic and basic drugs in ALS model cell lines, and find out the main transporters involved for the transport of those compounds. In our previously published work, we have used NSC-34 cell lines, which are also known as hybrid cell lines produced by the fusion of motor neurons in the spinal cord and neuroblastoma [22]. NSC-34/SOD1 ^G93A^, a mutant cell line (MT), has an overexpression of human SOD1 gene mutation due to the substitution at the 93 position of glycine with alanine [23]. NSC-34/SOD1 ^WT^ wild type (WT) is considered as the control. In our research work, we have compared the MT cell line with the WT cell line and together referred to them as ALS model cell lines. The procedure for the uptake in NSC-34 cells has been described in earlier studies [24,25]. Another cell model used in our previous research work is a BBB in vitro model. Various compounds have different structural properties [26], therefore it is of prime importance to study the transporters, which are able to mediate the permeability of drugs across blood to the brain [27]. BBB dysregulation causes increased permeability and leads to disease like Alzheimer’s disease (AD), ischemia and epilepsy. It is still unknown how the dysregulated BBB affects these various disorders. Animals served as the subject of early research on the BBB’s function in neurological illness and method for allowing the entry of medicinal substances [28]. Parallel artificial membrane permeability assay (PAMPA), which was developed by Kansy et al., has been studied for the transport of drugs to the brain [29]. However, PAMPA offers information regarding only passive diffusion, whereas it remains unaffected by the mechanisms including metabolism and active transport [30]. In our study, we have selected conditionally immortalized rat brain capillary endothelial cell lines (TR-BBB cells), which were established by harboring large T-antigen (temperature sensitive simian virus 40) from the transgenic rats [27]. The advantage of TR-BBB cell lines that possess the solute carrier transporters is that they help in determining the active delivery of drugs. The procedure for the culture and uptake the study has been descried earlier [31]. In addition, primary and immortalized brain microvascular endothelial cell lines (BMECs) have also been commonly used models for the study of drug delivery to brain [32].

Transporters for the amino acids have potential importance in the uptake of nutrients, signaling of cells, recycling of neurotransmitters, and expression of genes and maintain cell homeostasis [33]. The scope of the SLC transporters in brain and neurodegenerative diseases is broad, and beyond the scope of this paper; however, in this study, we intend to provide the overview of different transporters involved in the transport of amino acids, essential nutrients, and acidic and basic drugs in ALS model cell lines and TR-BBB cells. In this review, we have selected small amino acids such as alanine-serine-cysteine-threonine 1 (ASCT-1/Slc1a4) and alanine-serine-cysteine-threonine 2 (ASCT-2/Slc1a5) for the transport of l and d serine, respectively, taurine transporter (TauT/Slc6a6), large amino acid transporter 1 (LAT1/Slc7a5) for the transport of neutral amino acids for instance, and citrulline and cationic amino acid transporter (CAT1/Slc7a1) for the transport of basic amino acids such as arginine. In addition, we gave insight into the monocarboxylate transporters 1 (MCT1/Slc16a1) in the BBB, sodium-coupled monocarboxylate transporters (SMCT1/Slc5a7) in ALS for the transport of acidic drugs such as 4-phenylbutyrate (PBA), organic cationic transporters (OCTN1/Slc22a4 and OCTN2/Slc22a5) for the transport of carnitine in ALS cell lines, and OCTN2 for the transport of acetyl-l-carnitine (ALCAR) in TR-BBB cells. Choline transporter 1 (CHT/Slc5a8) and choline transporter-like protein-1 (CTL1/Slc44a1) in the BBB and ALS cell lines. Further, we have mentioned the effect of the inhibition of therapeutic drugs for ALS on the uptake rate of citrulline/LAT1 or valproic acid/SMCT1 transporters.

## 2. Slc1a4/ASCT1 and Slc1a5/ASCT2 Transporter in Motor Neuron Disease

As reported by an earlier study, ASCT1 is a potential transporter for l- and d- serine in astrocytes; in addition, it also acts as a shuttle for the transport of serine along the neuron and glia [34]. ASCT1 is marked as the obligatory exchange transporter and has advanced kinetics in comparison to the one-directional transporters for amino acids [35]. ASCT2, which is known as the transporter for neutral amino acids, also belongs to the SLC1 family, and it is found in numerous body sites localized in the plasma membrane [36]. An n-methyl-d-aspartate (NMDA) receptor co-agonist [37], d-serine, plays a role in several pathophysiological activities including neurotoxicity, neurotransmission and the formation of memory [38]. The relation between the d-serine and ALS was revealed by the findings of abnormal concentration levels of d-serine, which were shown in the mutant SOD1^G93A^ mouse and sporadic form of ALS [39,40]. The possible mechanism for the alteration in the level of serine has been reported as perhaps being deletion or mutation of the _d_-amino acid oxidase (DAO) gene [41]. For a clear understanding, the serine uptake in ALS model cell lines has been studied by Lee et al., 2017, and the findings suggest that the uptake of [^3^H]d-serine was markedly higher in the MT cells, whereas [^3^H]l-serine was higher in the WT cells. The kinetics parameters also elucidate the altered affinity and capacity. In case of [^3^H]d-serine, the affinity was higher in the MT cells as compared to WT; however, the affinity for [^3^H]l-serine in WT cells were two times lower in MT cells as compared to WT cells (Table 1). From the same study, it was revealed that ASCT1 was involved in the transport of [^3^H]l-serine, whereas ASCT2 transporter was involved for the transport of [^3^H]d-serine [42]. The altered levels of transporters were found via immunoblots in the transgenic ALS mice as compared to the non-transgenic mice [43].

## 3. Slc6a6/TauT Transporter in the BBB and ALS

Taurine transporter (TauT), a member of the SLC 6 family, and a sodium and chloride dependent transporter, Slc6a6, play an important role in taurine transport [51]. Taurine possesses osmoregulatory and antioxidant effects that help maintain homeostasis [52]. The brain controls the neuronal release of taurine in response to ischemia [53]. A previous study reported elevated taurine levels in the hippocampal slices under conditions of hypoglycemia and ischemia [54]. Our earlier study demonstrated that taurine transport activity at the BBB was reduced in hypertensive rats compared to the normotensive control rats [55]. Previous research has also shown that taurine, a sulfur-containing β-amino acid, plays an important role as a neuromodulator and neuroprotective agent against excitotoxicity and oxidative stress. Radiolabeled [^3^H]taurine is transported by TauT/Slc6a6 in rat brain capillary endothelial cells (TR-BBB13) [56]. Another research has shown that Slc6a6 uses GABA as a substrate, and this transport system seems to be present at the inner blood–retinal barrier [51]. The kinetic parameters of taurine have been studied in cultured bovine brain capillary endothelial cells (BCECs), and the [^3^H]taurine uptake has shown the activity of transporters at both membranes luminal and antiluminal of BCECs. Saturable taurine transport showed high affinity and low capacity systems, Michaelis-Menten constant (K_m,_ affinity); 12.1 ± 0.5 μM and velocity (V_max_); 4.32 ± 0.05 nmol/30 min/mg protein, for the luminal uptake, whereas, K_m_; 13.6 ± 2.4 μM and V_max_; 2.81 ± 0.02 nmol/30 min/mg protein, for antiluminal uptake of [^3^H]taurine [57].

The role of taurine in ALS has been demonstrated in a previous study. The immune reactivity of TauT was increased in the spinal cord of transgenic ALS mice (Male transgenic ALS mice are the MT SOD1 (G93A) expressing H1 high strain mice) in a pattern similar to that of the motor neurons of ALS patients [58]. Due to the increase in the taurine and TauT levels in the motor neurons of ALS, it was hypothesized it might be due to an increase in the uptake of taurine by the neurons. Therefore, the uptake study was performed in ALS model cell lines, and the data showed a time-dependent uptake of [^3^H]taurine, where the uptake was markedly higher in the MT cell line as compared to the WT cell (Figure 1). Further, the mRNA expression of TauT was higher in the MT cell line in comparison to WT [21].

Heat shock factor-1 (HSF1) mediated expression of TauT showed a compensatory effect against oxidative stress, which is considered the key factor in ALS pathogenesis. It has been concluded that TauT is one of the key markers for diagnosing stress in motor neurons, and the regulation of Slc6a6 may slow the process of neurodegeneration [21]. Additionally, the role of taurine was also studied against glutamate neurotoxicity, and the results showed that taurine protected the neurons from glutamate-induced injury; hence, taurine was considered valuable for use in ALS clinical trials [59].

## 4. Slc7a5/LAT1 Transporter in the BBB and ALS

The SLC7 family includes 15 members; two are pseudogenes, and the remaining 13 are divided into subgroups—the cationic amino acid transporters (CATs) and light subunits of LATs [60]. LAT1, associated with the SLC7 family, belongs to the amino acid-polyamine-organo cation (APC) superfamily [61]. SLC7a5 is responsible for the transport of amino acids and forms a heterodimer with glycoprotein SLC3A2 via a disulfide bond [62]. LAT1 is one of the important proteins responsible for the growth and development of cells because of its key role in the distribution of essential amino acids, especially in the placenta and BBB [63]. Neutral amino acids such as citrulline, a precursor of l-arginine [64], have been found to protect and prevent neuronal death and cerebrovascular injury. The role of citrulline in preventing cerebrovascular injury in the hippocampus was due to the regulation of endothelial nitric oxide (eNOS) [65]. Citrulline is transported in various cells such as neural cells [66], intestinal cells, macrophages [67], and bovine aortic smooth muscle cells [68] by different transport systems [44]. Citrulline delivery to the brain by LAT1 provides neuroprotection against cerebrovascular diseases. In the TR-BBB, the transport of [^14^C]citrulline through the BBB was carried by Slc7a5/LAT1 [44]. In addition, a previous study on ALS model cell lines reported that [^14^C]citrulline was mediated by Slc7a5/LAT1 transporter In a similar manner, the roles of essential amino acids such as tryptophan in both the BBB and ALS model cell lines have been studied, and the findings suggested that LAT1 was also involved in the transport of [^3^H]tryptophan [69].

A previous study on [^14^C]citrulline transport in the BBB showed that two saturable processes are involved in the transport, and the results showed that at high affinity site, higher affinity and capacity, whereas at a low affinity site, there was a lower affinity and capacity in the TR-BBB cell lines [44]. Another study has reported that in the BBB, LAT1 exhibits K_m_ values 1–10 μM for high affinity and 10–100 μM for low affinity [70]. Similarly, the kinetic parameters of [^14^C]citrulline in ALS have also been studied. The data showed that the high affinity and low capacity transport systems were involved in the MT compared to the WT [45]. These results are summarized in Table 1 and show that two affinity sites were involved in the BBB, whereas a single saturable process was involved in the ALS model cell line.

Reportedly, the hallmarks of ALS include oxidative stress and glutamate excitotoxicity [71]. Riluzole acts as a glutamate inhibitor and drug for the treatment of ALS. In addition, edaravone, known for its antioxidant effect, has recently been approved for ALS treatment [72]. Therefore, we aimed to compare the inhibitory effects of drugs on the SLC transporters such as LAT1 and SMCT1 substrate uptake from our previously published articles. In the ALS model cell lines, riluzole inhibited the uptake of [^14^C]l-citrulline in a concentration-dependent manner. A previous study reported no inhibition at 0.2 mM, whereas a significant inhibition was observed at 0.5 mM in both NSC-34 cell lines [45] (Table 2). Additionally, l-dopa, an l-system substrate and drug used for Parkinson’s disease, significantly inhibited citrulline uptake, showing the involvement of the LAT1 transporter in the transport of l-dopa in ALS cell lines. Furthermore, a previous study has shown in the Lineweaver–Burk plot analysis the competitive inhibition of citrulline with l-dopa. These findings indicated that l-dopa and citrulline compete for the same binding site, LAT1 [45].

## 5. Slc7a1/CAT1 Transporter in Motor Neuron Disease

The SLC7 family is subdivided as LATs and cationic amino acid transporters (CATs) [74]. For the transport of basic amino acids including lysine, arginine, and histidine Slc7a1 (CAT1) is mainly involved. Arginine has shown its potential role in the ALS by increasing the flow of blood resulting in the synthesis of protein and generation of α-ketoglutarate [75]. The scarcity of arginine makes neurons prone to excitotoxicity, and the addition of arginine has shown motor neuron protection against glutamte excitotoxicity [14]. A previous study on the transport of [^3^H]l-lysine has shown that CAT1 (system y+) was responsible for the transport of lysine in the ALS model cell line [24] and also across the BBB [76]. _L_-Arginine, a cationic amino acid, has important role in the pathogenesis of ALS [75] and has shown a potential role in enhancing the skeletal muscle growth and improving the glucose metabolic dysfunction [77]. The uptake of [^3^H]l-arginine was found to be concentration-dependent in ALS model cell lines, and the kinetics revealed that in the MT cell line, the affinity was lower and capacity was higher at a high affinity site, whereas at a low affinity site there was no significant difference between WT and MT cell lines, as shown in Table 1. According to the differential relative contribution study, it was shown that the system y+ (CAT1) mainly mediates the transport of [^3^H]arginine in ALS cell lines [46]. Similar patterns of results were shown in the inner blood–retinal barrier, showing the transport of arginine by carrier-mediated transporters [78]. Furthermore, the basic drugs, including quinidine, which is known for its antiarrhythmic actions, and verapamil, a calcium channel blocker, have shown the inhibitory effect of the transport of arginine in ALS model cell lines. Quinidine showed competitive inhibition with the Ki value of 0.64 mM in the disease model of ALS, showing that it competes with arginine for the same binding site and shared the same transporter, i.e., CAT1 in ALS model [46].

## 6. Slc16a1/MCT1 and Slc5a8/SMCT1 Transporters in BBB and ALS

MCTs play a vital role in cellular metabolism and energy pathways in several tissues [79]. The SLC16 family, which expands over 14 sub-members, is widely expressed in various organs such as the kidneys, heart, liver, adipose tissue, and brain [80]. In pathology and physiology, the commonly expressed and well-characterized members of the SLC16 family are Slc16a1/MCT1, responsible for the transport of pyruvates, ketones, and lactates, and Slc16a7/MCT2 and Slc16a3/MCT4 [81]. Genetic and metabolic studies have been linked to Slc16a1, and various mouse models have been developed to study the link between disease and transporter functions [82]. PBA, a short-chain fatty acid and histone deacetylase inhibitor, is involved in the treatment of various diseases [47]. Our previous study on the BBB has indicated the expression of MCTs, including rMCT1, 2, and 4. However, we observed that MCT1/Slc16a1 was the main transporter in [^14^C]PBA transport to the brain across the BBB [47]. Additionally, [^14^C]PBA transport characteristics and transporters involved have been studied in ALS cell lines, and the results indicated that sodium-coupled MCT1 (SMCT1/Slc5a8) and MCT1 both help [^14^C]PBA transport to NSC-34 cell lines [48].

[^14^C]PBA transport by TR-BBB showed that the transport was concentration-dependent, and the Michaelis–Menten constant demonstrated that the carrier-mediated transport of PBA was pH-dependent, with the K_m_ four times higher at pH 7.4 than the K_m_ at pH 6.0. In contrast, V_max_ was five times lower at pH 7.4 than pH 6.0 (Table 1) [47]. In addition, transport kinetics of [^14^C]PBA in ALS model cell lines showed the two affinity sites with altered affinity and capacity. At the high-affinity site, the capacity was five times lower in the MT than in the WT, whereas, at the low-affinity site, affinity was three times lower in the MT than in the WT [48] (Table 1).

Conversely, our previous study on valproic acid (VPA) has shown the neuroprotective effects of VPA in the ALS disease model. The study data suggested that the transporter SMCT1 was commonly involved in mediating the transport of VPA in NSC-34 cell lines [73]. Transport of [^3^H]VPA in ALS cell lines was concentration-dependent, and the saturation kinetic parameters demonstrated two affinity sites. MT possessed significantly higher affinity and capacity than the WT at the high-affinity site, whereas, at the low-affinity site, MT showed lower capacity than WT. In the brain endothelial and intestinal epithelium the affinity for VPA ranged between 0.6–0.8 mM [83], that is likewise the K_m_ value in NSC-34 cell lines. Other SMCT carried monocarboxylates exhibited the K_m_ value between 0.07–6.5 mM that is also comparable to motor neuronal cell lines [84]. Ibuprofen, a strong inhibitor of SMCT1 and an anti-inflammatory agent, significantly inhibited [^3^H]VPA uptake at the concentration of 10 mM in WT and MT (Table 2). In addition, PBA, a substrate of SMCT1, also strongly inhibited the transport of VPA in ALS cell lines, suggesting that they both utilize the same transporter system, SMCT1. Furthermore, edaravone, an organic anion transporter (OAT) substrate and a drug for ALS treatment, significantly inhibited [^3^H]VPA uptake up to about 68% inhibition at 10 mM in both cell lines [73] (Table 2). A previous study on half maximal inhibitory concentration (IC_50_) analysis in MT revealed that high edaravone concentration is required to reach 50% inhibition [73]. These results indicated that drugs like ibuprofen, PBA, and edaravone possibly use the transporter SMCT1 and inhibit [^3^H]VPA uptake.

## 7. Slc22a4/OCTN1 and Slc22a5/OCTN2 in BBB and ALS

Lee et al., 2012, have studied the transport properties of acetyl-l-carnitine (ALCAR) in the BBB [49]. In the brain, kidney, liver and intestine, ALCAR is produced from carnitine and acetyl coenzyme A. Various physiological effects of ALCAR have been studied in the brain mainly, where ALCAR helps in the transmission of numerous neurotransmitters, morphology of synapsis, brain energy modulation and as a neurotrophic factor [85]. It has been shown that [^3^H]ALCAR transport in TR-BBB cells was carried by OCTN2. Expression of OCTN2 in the cells confirm the involvement of OCTN2 in TR-BBB cells. Another study has reported that OCTN2 in the brain and astrocytes are responsible for the transport of ALCAR and l-carnitine [86]. The kinetic parameters from the concentration dependent uptake study in TR-BBB cells showed that a single transport system is involved for the uptake of ALCAR (Table 1). Earlier research on l-carnitine has shown the involvement of both OCTN1 and OCTN2 in the transport of carnitine in motor neuron NSC-34 cell lines [25]. Another study has shown that the administration of the energy metabolizing entity l-carnitine to neuronal cells in human has increased neuronal mitochondrial functions and thus has a role in preventing the pathological conditions related to ALS disease [87]. The concentration-dependent uptake of [^3^H]l-carnitine in ALS model cell lines exhibits saturable processes and two affinity sites. The data shown in Table 1 indicated that in MT, the cell affinity is higher and the capacity is lower, which is significantly different from the WT cell line at a low affinity site [25]. An animal study of juvenile visceral steatosis (JVS) disease mice has shown the reduced transporter activity due to the reduced capacity, which supports the findings in NSC-34 cell lines [88]. Additionally, the study in neural cells have shown the high affinity for carnitine transporters [89]. From Table 1, it is concluded that ALCAR exhibit a single affinity site in TR-BBB cells, whereas l-carnitine possesses two affinity sites in ALS model cell lines. Additionally, various pharmacological compounds such as quinidine, pyrilamine, diphenhydramine (DPH) and metformin have shown the significant inhibitory effect of the uptake of carnitine in ALS model cell lines. The organic cationic compounds showed competitive inhibition with l-carnitine, showing they compete for the same binding sites [25]. In addition, it has been reported that the OCTN2 transporter showed a high affinity, whereas OCTN1 showed a low affinity for the carnitine [90]. A human study has shown that the mutations in OCTN2 results in a deficiency of carnitine and resulted in muscle weakness and cardiomyopathy [91]. Furthermore, l-carnitine showed significant inhibition on the uptake of [^3^H]paeonol in ALS model cell lines. The OCTN1 and PMAT transporters showed altered behavior in the disease model of ALS in the uptake of paeonol [92].

## 8. Slc5a7/CHT in the BBB and Slc44a1/CTL1 in Motor Neuron Disease

Choline is an essential nutrient and hydrophilic cationic compound for plasma cell membrane synthesis [93]. It is an important neurotransmitter for cholinergic neurons that release acetylcholine (ACh) for the sympathetic and parasympathetic systems [94]. For the uptake of choline, various transport systems are involved depending on the affinity of choline. A high affinity, hemicholinium-3, and sodium dependent choline transporter (CHT1) has the rate limiting role for the synthesis of ACh [95]. Mutations in this transporter result in neurological diseases including depression and AD [96]. Another intermediate affinity transporter include choline-like transporters (CTLs). CTL1 is a member of the broader Slc44a1-5 family [97]. Choline transport for membrane phospholipids synthesis is carried by CTL1/Cdw92/Slc44a1 and is considered a major contributor to the family [98]. Reportedly, the homologous CTL1 genes were found in rats, mice, and humans [99,100,101]. In the mitochondria and plasma membrane, CTL1 is a choline /H^+^ antiporter [102]. The exact role and function of CTL2/Slc44a2 are not well known; however, it is indirectly involved in phosphatidylcholine synthesis [103]. Choline scarcity affects various processes, including the expression of genes involved in cell differentiation, apoptosis, and proliferation. In addition, low affinity organic transporters (OCTs) are also responsible for the transport of choline [104]. Research has shown that OCT1 as well as OCT3 expression increases the uptake of choline in in Xenopous oocytes **[105]**. The abnormal metabolism and transportation of choline are involved in neurodegenerative disorders like PD and AD [98]. In hypertensive rats, the alteration in the function of the choline transport system has been reported, and the change in choline transport activity is of prime physiological importance as the brain is incapable of producing choline de novo [104].

A previous in vivo choline study via a carotid artery injection and isolated brain capillaries via an in vitro technique has shown that choline transport was implicated by a carrier-mediated system in the BBB. Our previous study in the rat conditionally immortalized syncytiotrophoblast cell line (TR-TBT) provided an analysis of the various choline transporters, and the results demonstrated CTL1 expression in TR-TBT cells. In addition, CHT and CTL1 were expressed in the rat brain and placenta (Figure 2A) [93].

Further, in our previous study on TR-BBB cell lines, CHT1 was expressed in the brain, whereas OCT2 was expressed in the brain and TR-BBB (Figure 2B) [50]. CTL is the main transporter for the transport of choline in ALS model cell lines and showing the relative lower expression of CTL1 in MT as compared to WT. Additional studies have reported that choline transport was carried by CTL1 in the mouse neurons primary cultures and rat astrocytes and carcinoma lung cells [106,107].

Reportedly, choline uptake varies due to the differences in the transporter family involved and its sodium dependency and distribution in tissues [108]. Further, the kinetic parameters of [^3^H]choline transport in the BBB were retrieved from our previously published article. The data presented in Table 1 indicated that carrier-mediated saturable processes are involved in the uptake rate of choline in the BBB. At the high affinity site, the affinity was higher and the capacity was lower as compared to the lower affinity site (Table 1) [109]. In TR-BBB cell lines, in vitro [^3^H]choline uptake was concentration-dependent, and the Eadie–Hofstee plot showed a straight line, indicating a single saturable process with K_m_ (26.2 µM) and V_max_ (397 pmol/mg protein/min) [50]. In addition, another study demonstrated that choline transport in the brain and spinal cord was carried by CHT1 in a sodium-dependent manner and possessed K_m_ ranging from 0.5 to 3 µM [110]. Furthermore, an organic cation, choline, in multiple tissues has been reported to be transported by CTL1 in a sodium-independent manner, and the K_m_ values varied from 10–50 µM [111].

A previous study on the transport of choline through TR-BBB cell lines has illustrated the inhibitory effect of basic drugs, including choline, hemicholinium-3, a choline analog, and α-phenyl-n-butyl nitrone (PBN); the study results showed that these drugs inhibited choline uptake in TR-BBB cells and exhibited IC_50_ of 9.40 µM, 37.2 µM, and 1.20 mM, respectively (Figure 3A). In ALS model cell lines, the inhibitory effect of pharmacological drugs, including amiloride and DPH, was studied. IC_50_ was calculated as 1.04 µM for amiloride, and 61.0 µM for DPH. These drugs showed inhibition in a dose–response manner in the MT cell line (Figure 3B).

Lower concentrations varying from 0.001 to 3 mM of these drugs were required to show inhibition of [^3^H]choline uptake by half in the ALS disease model cell lines. Amiloride was the most sensitive drug in the ALS model cell line to achieve the maximal half concentration, whereas choline inhibition showed the most sensitivity in the TR-BBB cell line. A previous study on muscle cell lines involving the H1 receptor antagonist, DPH, showed the inhibition of histamine receptors, and the estimated IC_50_ value, 1.01 µM, helped control the intracellular calcium [112]. Another study reported the IC_50_ of amiloride for the inhibition of calcium channels as 30 µmol/L in the mouse neuroblastoma [113]. It must be noted that the inhibitory concentrations varied according to the cell type and concentrations.

## 9. Concluding Remarks

Overall, in this review, the representative transporters, which are related with the transport of amino acids including small amino acid, large neutral amino acids, basic amino acids, as well as acidic drugs including PBA and VPA and essential nutrients such as choline and carnitine in BBB and ALS model cell lines. Affinity and capacity plays important role in understanding the transport ability of compounds into the cells. Therefore, we compared the kinetic parameters obtained in BBB and ALS cell lines. In the BBB, affinity of LAT1 is very high, but capacity is very lower than ALS cell lines. l-Arginine transported by CAT1 in ALS cell lines, showed low affinity and higher capacity in MT cell line. Taurine levels were altered in the transgenic ALS mice, in addition to this, the uptake of [^3^H]taurine was higher in the disease model as compared to the control. It was concluded that increase in the levels of taurine might be due to increase in the TauT levels in the disease model cell line. TauT modulation might delay the neurodegeneration and is considered to be a novel biomarker for ALS. PBA was transported across the BBB by MCT1; however, in addition to MCT1, PBA transport in ALS cell lines was mediated by SMCT. _L_-Carnitine was transported by OCTN1/2 in ALS cell lines whereas its acetylated form ALCAR transport was mediated by OCTN2 in TR-BBB cells. CHT1 was the main transport for choline across the BBB; however, no CHT1 expression was observed in ALS cell lines. CTL1 is responsible for choline transport in NSC-34 cell lines. In addition, IC_50_ evaluation of basic drugs showed that choline in BBB and amiloride were the most sensitive in ALS cell lines. Additionally, it is shown that the therapeutics for ALS such as riluzole and edaravone can be transported by LAT1 or SMCT1, respectively, in ALS cell lines. In conclusion, ALS is a devastating neuronal disease and need of hour is to find the possibilities to cure and alleviate the symptoms related to ALS. Hence, adequate knowledge of transporter involvement will be beneficial in delivering novel drugs in ALS. Comprehending the concept of SLC transporters involved in the transport of compounds will aid the development of new drugs and their delivery in brain and motor neuron diseases.

## Figures and Tables

**Figure 1 pharmaceutics-14-02167-f001:**
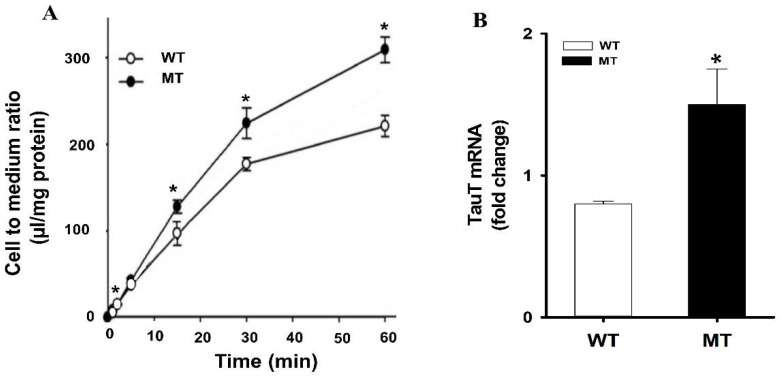
Reproduced from Jung et al., Molecular Neurobiology, 2013 [21]. (**A**) Uptake of [^3^H]taurine in a time dependent manner. Uptake was carried at temperature 37 °C. (**B**) Relative expressions of TauT transporter in ALS model cell lines. mRNA expression levels were determined using quantitative RT-PCR analysis and normalized to the internal control GAPDH in ALS model cell lines. Each value represents the mean ± SEM. (*n* = 3–4). * *p* < 0.05 indicates a significant difference with respect to the WT control.

**Figure 2 pharmaceutics-14-02167-f002:**
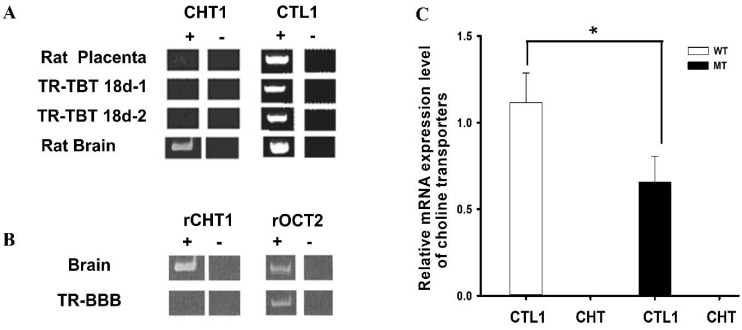
Relative expressions of transporters in ALS model cell lines. The data in (**A**) was retrieved and analyzed from the earlier published study by Lee et al., Placenta, 2009 [93]. (**B**) Determination of rCHT1 and rOCT2 in TR-BBB cells. This data is retrieved from our previous research by Kang et al., Archives of Pharmacal Research, 2005 [106]. (**C**) CTL1 mRNA expression levels were determined using quantitative RT-PCR analysis and normalized to the internal control GAPDH in ALS model cell lines. Each value represents the mean ± SEM. (*n* = 3–4). * *p* < 0.05 indicates a significant difference with respect to the WT control.

**Figure 3 pharmaceutics-14-02167-f003:**
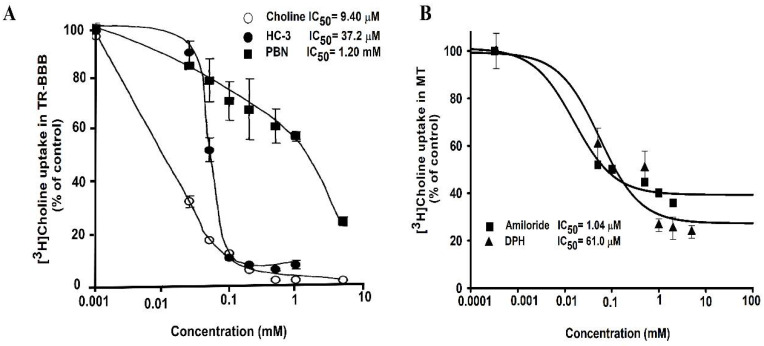
Inhibitory effect of drugs in a dose–response manner. (**A**) Inhibition of choline uptake inhibition by choline, hemicholinium-3, and α-phenyl-*n*-butyl nitrone (PBN) in TR-BBB cells. These data were retrieved and analyzed from the earlier published study by Kang et al., Archives of Pharmacal Research, 2005 [106]. (**B**) Half inhibitory concentrations (IC_50_) were analyzed either in the presence or absence of diphenhydramine (DPH), and amiloride at concentrations of 0–2 mM at pH 7.4 and 37 °C on [^3^H]choline uptake in the ALS disease model cell line (MT). Data are represented as the mean ± S.E.M (*n* = 3–4).

**Table 1 pharmaceutics-14-02167-t001:** Kinetic parameters analysis of various transporters in BBB and ALS model cell lines.

Transporters (Substrate)	Affinity (mM)			Velocity (nmol/mg Protein/min)		
	BBB	WT	MT	BBB	WT	MT
ASCT1 (_L_-Serine) ^a^	--	0.061 ± 0.004	0.0308 ± 0.0021	--	1.94 ± 0.01	1.69 ± 0.07
ASCT2 (_D_-Serine) ^a^	--	11.3 ± 1.3	21.1 ± 3.0	--	39.5 ± 1.4	41.5 ± 2.0
LAT1	0.031 ± 0.001	1.48 ± 0.21	0.670 ± 0.050	0.185	18.3 ± 2.9	10.9 ± 0.8
(Citrulline) ^b,c^	0.0017 ± 0.0004	--	--	0.0032	--	--
CAT1	--	0.013 ± 0.005	0.30 ± 0.11 ***	--	0.012 ± 0.006	0.47 ± 0.15 **
(Arginine) ^d^		3.51 ± 1.73	1.98 ± 1.10	--	3.30 ± 1.62	1.42 ± 1.30
MCT1 (PBA) ^e^	13.4 ± 2.9	--	--	4.16 ± 0.55	--	--
SMCT1 (PBA) ^f^	--	0.514 ± 0.068	0.314 ± 0.031	--	0.562 ± 0.035	0.109 ± 0.046
		2.66 ± 0.19	7.69 ± 0.44	--	2.66 ± 0.19	4.17 ± 1.38
OCTN2 (ALCAR) ^g^	0.054 ± 0.009	--	--	1.07 ± 0.05	--	--
OCTN1/2 (Carnitine) ^h^	--	0.0019 ± 0.0003	0.0020 ± 0.0003	--	0.00030 ± 0.0001	0.00019 ± 0.00003
	--	0.994 ± 0.034	0.374 ± 0.089 ***	--	0.259 ± 0.009	0.062 ± 0.013 ***
CHT (Choline) ^i^	0.020	--	--	0.019	--	--
	0.210	--	--	0.167	--	--

The kinetic parameters of various transporters in the BBB and ALS model cell lines. ^a,b,c,d,e,f,g,h,i^ These data points were retrieved from the previously published articles [25,42,44,45,46,47,48,49,50]. In front of each transporter, the upper row shows the high affinity site, and the lower row represents the low affinity site, respectively. ** *p* < 0.01, and *** *p* < 0.001 represent significant differences from the respective WT.

**Table 2 pharmaceutics-14-02167-t002:** Inhibition effect of therapeutics of ALS on the uptake rate of transporters substrate in ALS model cell lines.

Drugs (% of Control)	Conc. (mM)	LAT1 (^14^C-Citrulline) ^a^		SMCT1 (^3^H-VPA) ^b^	
		WT	MT	WT	MT
+_L_-Dopa	0.5	32.8 ± 2.7 ***	36.8 ± 12.0 ***	--	--
+Riluzole	0.5	72.6 ± 3.4 **	80.0 ± 1.2 **	--	--
+Ibuprofen	10	--	--	22.3 ± 1.5 ***	33.5 ± 1.0 ***
+PBA	10	--	--	43.1 ± 2.2 ***	56.6 ± 3.0 ***
+Edaravone	10	--	--	67.3 ± 7.6 **	68.6 ± 6.3 **

The percentage of inhibition on each transporter substrate uptake of drugs in NSC-34 cell lines. ^a,b^ Data points were taken from our earlier published articles [45,73]. ** *p* < 0.01, and *** *p* < 0.001 represent significant difference from the respective control. VPA (valproic acid), PBA (4-phenylbutyric acid).

## Data Availability

All data were included in this study.

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
