# Peer review of "Blood–Brain Barrier Solute Carrier Transporters and Motor Neuron Disease"

_pharmaceutics, 2022, doi:10.3390/pharmaceutics14102167_

Round 1

Reviewer 1 Report (New Reviewer)

The subject of solute carrier transporters in the CNS is very interesting and deserves a thorough and clear review. The Kang lab obviously has the experience and the expertise necessary in the field to write such a review. However, this is not the case in this manuscript that is a strange mix between regular article and review article and essentially doesn’t fit either category. Although it is presented as a regular Article, I assume that it is supposed to be a review since the data presented is not new. As a review, it doesn’t adhere to good Review practice rather it mainly summarizes data obtained and published in the lab about SLC transporters in 2 types of cells. To be published as a review, the authors need to review the field in depth while giving a larger scope with emphasis and a clear connection between BBB and ALS as well as differences between species, animals, cells, human studies etc. What does the affinity and capacity data mean and how it is compared to the literature. There are more issues to be addressed but this paper should be first completely revised and submitted as a proper review article and not as a summary of results obtained in the authors lab.   

Author Response

Reviewer 1

The subject of solute carrier transporters in the CNS is very interesting and deserves a thorough and clear review. The kang’s lab obviously has the experience and the expertise necessary in the field to write such a review. However, this is not the case in this manuscript that is a strange mix between regular and review article and essentially doesn’t fit either category. Although it is presented as a regular article, I assume that it is supposed to be a review since the data presented is not new. As a review, it doesn’t adhere to good review practice rather it mainly summarize data obtained and published in the lab about SLC transporters in 2 types of cells to be publish as a review, the authors need to review the field in depth while giving the larger scope with emphasis and a clear connection between BBB and ALS as well as the differences between species, animals, cells and human studies etc. what does the affinity and capacity data mean and how it is compared to the literature. There are more issues to be addressed but this paper should be first completely revised and submitted as a proper review article and not as a summary of the results obtained in the author’s lab.

Response:

According to the reviewers comment, we have changed the manuscript as a review article. We have summarized the characteristics of the most representative transporters and their role in the transport of the essential nutrients, acidic drugs and basis drugs in BBB and ALS model cell lines. This study help significantly to understand the concept of SLC transporters (comparison with affinity and capacity of transporters) involved in the transport of various compounds, being a step stone to the new drugs development process and their associated delivery process in brain and motor neuron diseases. We have modified abstract Line # 12-13 and # 19-21

In section 1( Introduction), we have added description in Line # 40-41, # 90-101, and #110-111.

In section 2 (Slc1a4/ASCT1 and Slc1a5/ASCT2 Transporter in Motor Neuron Disease) Line # 148-149.

In section 5 (Slc7a1/CAT1 Transporter in Motor Neuron Disease), Line # 258-263.

In section 6 (Slc6a1/MCT1 and Slc5a8/MCT1 Transporter in BBB and ALS), Line # 303-306.

In section 7 (Slc22a4/OCTN1 and Slc22a5/OCTN2 in BBB and ALS), Line # 344-347, and #349-358.

Reviewer 2 Report (New Reviewer)

The article is well designed and written. The study seems interesting, it could be accepted due to significance and further advantages compared to the existing literature.

I have only one substantive objection,

I think that the work should mention the use of the Pampa BBB model

Author Response

Reviewer 2

The article is well designed and written. The study seems interesting, it could be accepted due to significance and further advantages compared to the existing literature.

I have only one substantive objection. I think that the work should mention the use of the Pampa BBB model.

Response:

We are thankful to the reviewer for such encouraging comment. We have added the description of Pampa BBB model in our manuscript in introduction section highlighted in yellow color, Line # 90-98.

Reviewer 3 Report (New Reviewer)

The authors of this review have summarized the characteristics of the most representative transporters and their role in the transport of the essential nutrients and acidic drugs in BBB and ALS model cell lines.

Overall, this work will help significantly to understand the concept of SLC transporters involved in the transport of various compounds, being a step stone to the new drugs development process and their associated delivery process in brain and motor neuron diseases.

Author Response

Reviewer 3

The authors of this review have summarized the characteristics of the most representative transporters and their role in the transport of the essential nutrients and acidic drugs in BBB and ALS model cell lines. Overall, this work help significantly to understand the concept of SLC transporters involved in the transport of various compounds, being a step stone to the new drugs development process and their associated delivery process in brain and motor neuron diseases.

Response:

We are thankful to the reviewer for such encouraging comment.

Round 2

Reviewer 1 Report (New Reviewer)

Can be accepted

This manuscript is a resubmission of an earlier submission. The following is a list of the peer review reports and author responses from that submission.

Round 1

Reviewer 1 Report

If this is designed as a review on the role of SLC's in ALS it should give the summary of the literature on the subject. Several SLC's involved in ALS are not addressed (SLC1A2, 1A1, 25A20, 57...). 

More information on the cells and models used would be necessary to understand the work.

Reviewer 2 Report

This Review article from Latif and Kang is concerned with the description of several solute carrier transporters in the blood brain barrier and in motoneuronal-like NSC-34 cells. Most of the data presented relate to previous experimental research done by the authors' group. There is no clear logic as to how the information is presented or what information is included. The manuscript is full of inaccuracies and mistakes, such as the description of ALS as mostly caused by excitotoxicity and oxidative stress, missing references (e.g. lines 136-138 and lines 168-170), and wrong references listed (e.g. ref 47 and ref 15). The figures presented are taken from previous studies, but the data is not properly explained. For instance, in Figure 1 data regarding CHT1 and OCT2 is shown,  but these two transporters were not previously introduced in the text. Table 1 is referred to on page 5, line 179, but the data described in that paragraph is not in any of the tables included. Table 1 cited on line 183 should be Table 2. Finally, the manuscript is hard to read and would require extensive English editing.